# Comparing Analogy-Based Methods—Bio-Inspiration and Engineering-Domain Inspiration for Domain Selection and Novelty

**DOI:** 10.3390/biomimetics9060344

**Published:** 2024-06-06

**Authors:** Sonal Keshwani, Hernan Casakin

**Affiliations:** 1Human Centered Design Indraprastha Institute of Information Technology, Delhi 110020, India; sonal.keshwani@iiitd.ac.in; 2School of Architecture, Ariel University, Ariel 40700, Israel

**Keywords:** bio-inspired design, transformative solutions, design creativity, conceptual design, Design-by-Analogy

## Abstract

This study aims to support designers in developing transformative solutions in the engineering discipline using the Design-by-Analogy ideation method. Design-by-Analogy involves drawing inspiration from the source domain and applying it to the target domain. Based on the conceptual distance between the two domains, analogies are classified as biological—(natural), cross—(distant-engineering), and within—(near-engineering) domain analogies. Real-world scenarios involve designers selecting analogies after seeking them across multiple domains. These selected analogies significantly influence the produced designs. However, the selection criteria of the analogy domain are unexplored in design research. We address this gap by investigating: (a) the influence of analogy domains on their selection frequency; and (b) the relationship between the frequency of selecting analogies from specific domains and the novelty of designs. The experiment involved twenty-six teams of novice product designers, who solved design problems aided by one analogical source from each domain. The results showed that biological analogies were frequently selected. While biological-domain analogies significantly increased the novelty of designs compared to the within-domain ones; no significant difference was found between the biological- and cross-domain analogies, suggesting that middle-domain analogies can be as effective as far-domain ones. The findings can support technological innovation by aiding the development of analogy search databases.

## 1. Introduction

This study aims to assist organizations in producing highly novel designs—a central aspect of transformative solutions. Supporting novelty in designs is essential as it positively influences the organizations’ market share [1,2]. In design literature, ‘novelty’ refers to a product’s originality or newness concerning the existing products in the market [3,4,5,6]. The invention of the first X-ray machine, the first pinhole camera, and the first wheel are some examples of highly innovative products [6].

When aiming to produce novel designs, designers use methods to break away from traditional ideas. A commonly used design method is Design-by-Analogy (DbA) [7,8,9,10]. In this method, the designers take inspiration from a familiar domain (referred to as the source domain) to solve the problem in an unfamiliar domain (referred to as the target domain) [11]. In the product design context, the target domain is typically engineering. A classic example of problem-solving using DbA is the adaptation of astronauts’ vortex cooling systems for cooling molds and glass bottles [12]. Several researchers have focused on studying the influence of the conceptual distance between the source and the target domains on the novelty of designs [13,14,15,16,17,18,19]. However, the selection criteria of the analogy domain are largely unexplored in design research. This study addresses this gap by investigating: (a) the influence of analogy domains on their selection frequency; and (b) the relationship between the frequency of selecting analogies from specific domains and the novelty of the design solutions. The results of this work can provide theoretical guidance for the development of tools for analogy search. Based on the findings, these tools can be modified to prioritize the search results from those analogy domains that are more likely to be selected and are expected to produce more novel designs. This, in turn, will support technological innovation.

The remaining parts of this section present the literature on the DbA method and outline the research questions, objectives, and hypotheses addressed in this work.

Several classification schemes have been proposed for analogy domains (see Table 1). Firstly, analogies were classified as far-domain and near-domain based on the conceptual distance between the source and the target domains [8,9,15,18,20]. Researchers modified this bipartite classification into a tripartite classification by proposing middle-domain analogies relative to far-domain and near-domain analogies [13,14,17]. Table 2 shows examples of far-, middle-, and near-domain analogies [13]. Secondly, based on engineering and biology disciplines, analogy domains were classified into engineering-domain and biological-domain analogies, respectively [21,22]. This bipartite classification was converted to the tripartite system of analogies: near-engineering (in-domain), far-engineering, and biological domain [16]. Table 1 shows the influence of this classification on the novelty of designs.

Overall, the plethora of studies on the relationship between the two parameters—‘analogy domains’ and ‘novelty of designs’—illustrates the importance of selecting analogous systems from those domains that are perceived to support generating design ideas with larger novelty.

In the process of ideation, designers encounter multiple analogies. While these analogies can be recalled from memory [24], tools such as Asknature [25] support designers in searching relevant biological analogies. A keyword search in Google is another widely used method for analogy search [26]. Amongst the myriad analogies displayed as search results by these tools, designers ‘select’ just a few. As the selected analogy influences the design outcome, it is necessary to understand how the domain influences the designers’ selection of certain analogies over others. Therefore, this study focuses on how domain influences the designers’ selection of certain analogies over others.

Previous studies [26,27,28] used a protocol analysis approach to categorize the reasons behind the selection of an analogy. These are function, form, originality, symbolism, aesthetics, design process, nature, structure [28] and experience, physical property, and feeling [27]. In addition to this, Chai et al. [27] compared, between the experts and novice designers, the influence of distant-, medium-, and near-domain analogies on analogy selection. They found that experts chose near-domain analogies while novice designers chose distant-domain analogies. Along similar lines, Lu et al. [26] compared designers’ and non-designers’ selections of analogies belonging to near-, middle-, and distant-domains that were searched using Google Image Search. These authors found that designers selected middle-domain analogies, whereas non-designers selected distant-domain analogies.

While the influence of near-, middle-, and far-domain selection has been studied, to our knowledge, no work investigated the analogy selection from the biological domain. One related work is that of Ruiz-Pastor et al. [29], who compared two types of stimuli—random and biomimetic stimuli—regarding creativity and circularity of the produced concepts. Our work differs from their work in the following ways:Ruiz-Pastor et al. [29] have considered two types of stimuli—random stimuli and biomimetic stimuli. However, our work considers three types of analogy domains (stimuli): biological, cross, and within—both cross and within belonging to the engineering domain. They do not specify the discipline used for random stimuli; on the contrary, we employ the engineering discipline for cross- and within-domain analogical sources.They [29] exposed the designers to nine random images and nine biomimicry cards for random stimuli and for biomimetic stimuli, respectively. The random images and biomimicry cards were given to the designers in two separate sessions, and the designers were free to select any random image or any biomimicry card. While this experiment allowed the designers to select inspiration sources from each type of stimuli, it did not allow them to make selections across the different types of stimuli considered—something that we intended to do in our work (see Section 2.1 for details).While [29] compared the influence of the two stimuli on novelty, they have not studied the influence of the selection frequency of stimuli on novelty—which is one main goal of our work.

Bearing that there has been an exponential growth in the use of biological-domain analogies [30,31,32,33,34], it is essential to understand why designers prefer to select biological-domain analogies over analogies from the other two domains. We address this gap by choosing the classification scheme proposed by Keshwani et al. [16], i.e., biological-domain, cross-domain, and in-domain analogies (hereafter referred to as within-domain in this work) to study its influence on analogy selection.

Hence, the first research objective is to understand the influence of biological-, cross-, and within-domain analogies on their selection frequency during the design process. Based on the results of Lu et al. [26], who reported that designers select middle-domain analogies, we hypothesize the following (H1):The **frequency of selecting** cross-domain analogies will be higher than that of the biological- and within-domain analogies;Following the path of least resistance [20], within-domain analogies will have a higher **selection frequency** than biological-domain analogies.

Thus, hypothesis H1 is mathematically expressed as f_cross_ > f_within_ > f_bio_.

As the conceptual distance between the source and the target domain influences the novelty of designs [13,14,15,16,17,18,19], the second research objective is to understand the relationship between the frequency of selection of analogy domains and the novelty of the design solutions. The frequency of selection of analogy domains also represents fluency of ideas produced using the selected analogy.

Previous studies reported that far-domain analogies produced more novel designs than near-domain analogies [8,15]. As biological-domain analogies are the farthest from the target domain, followed by cross- and then within-domain, we formulate the second hypothesis (H2): the **strength of the correlation** (r) between the **overall novelty of ideas** (n) produced in a design session and the **frequency of domain ‘d’ selection** (f_d_) will vary in the following order: r_bio_ > r_cross_ > r_within_.

To our knowledge, no study has compared the influence of analogy domains on the correlation coefficient between novelty and the selection frequency, which, in our work, is considered the fluency of ideas.

Table 3 summarizes the research questions, objectives, and hypotheses, where:

f_d_ is the ‘frequency f of selection of analogies from domain d’;

n is the ‘overall novelty of ideas produced by a design team in the design session’;

r_d_ is the strength of the correlation (r) between the ‘novelty of ideas produced’ (n) and ‘frequency of selection of domain d’ (f_d_).

The remaining sections of this paper are organized as follows: Section 2 provides detailed information and an analysis of the experiment conducted. Section 3 presents the method adopted to analyze design ideas. Section 4 presents the results, followed by Section 5 discussing them. Lastly, Section 6 focuses on the conclusions drawn from this work.

**Table 3 biomimetics-09-00344-t003:** Research questions, objectives, and hypotheses.

Research Question	Research Objective	Research Hypothesis
RQ1: What is the influence of biological-, cross-, and within-domain analogies on their selection frequency (f_d_) during the design process?	O1: To understand the influence of biological-, cross-, and within-domain analogies on their selection frequency (f_d_) during the design process.	H1: f_cross_ > f_within_ > f_bio_
RQ2: What is the relationship between the frequency of selecting analogies from specific domains (f_d_) and the novelty of the design solutions (n)?	O2: To understand the relationship between the frequency of selecting analogies from specific domains (f_d_) and the novelty of the design solutions (n).	H2: r_bio_ > r_cross_ > r_within_

## 2. Materials and Methods

This section describes the empirical approach adopted to test the hypotheses. Section 2.1, Section 2.2, Section 2.3 and Section 2.4 present the procedure details, subjects, experimental setup, and materials, respectively.

### 2.1. Procedure

An experiment was designed involving 26 teams, each composed of two members. Each team was allocated one design problem and three analogies—one from each domain—biological, cross, and within. Subjects were asked to generate as many design concepts as possible for the given design problem, with each idea being inspired by any of the three analogies. No restriction was placed on the sequence of analogies used for idea generation. This experimental design aimed to emulate a more realistic scenario where designers encounter analogies belonging to different domains and select some based on specific criteria. The subjects were instructed to present their design ideas as annotated sketches, and each session lasted 40 min. The objective of the experiment was not revealed to the subjects to prevent biases in analogy selection. Moreover, using the Internet and mobile phones was prohibited during the experiment.

### 2.2. Subjects

Fifty-two students from the second year of the Master’s program in Design and the first and second years of the doctoral Design program at the University Design Department were selected for the experiment. All the subjects had Bachelor’s in Mechanical or Production Engineering. Out of these, 45 were males, and 7 were females, with an average age of 24. Students were from central and southern geographical regions of India, such as Karnataka, Maharashtra, Andra Pradesh, Tamil Nadu, and Kerela. However, all of them were Indians with a similar urban background. This variability in design education, gender, and origin was due to the availability of the participants with the required educational background at the time of the experiment.

Twenty-six teams were formed, each consisting of two subjects. This setup allowed the subjects to externalize their thoughts verbally while designing. Each team was assigned a name from A to Z. As the subjects had a background in design, they were familiar with the DbA method.

### 2.3. Experimental Setup

The experiment was conducted in Bengaluru, in the University Design Department. We chose a multi-media classroom with a seating capacity of one hundred. The seating arrangement was adjusted to facilitate face-to-face discussions between the subjects within each team. Teams were located sufficiently far apart so that other teams could not overhear the discussions. To eliminate the influence of unintended external stimuli on the design outcomes, the room lacked any image or graphic display on the walls. The researchers provided instructions to the participants before the commencement of the design sessions. To keep consistency, all instructions were printed and provided to each team (see Appendix A). Prior consent was obtained from the subjects to allow the data generated from the experiment to be used for research purposes.

### 2.4. Materials

The design problem was presented as follows: Modern houses have underground sumps where residents can store water for future use. Over time, however, contaminants, such as insects, dust particles, and mosquitoes, find their way into the sump, rendering stored water unfit. Cleaning these sumps is challenging. Some dirty water inevitably remains inside and cannot be removed efficiently. Due to the depth of the sumps, manual removal of water and debris becomes impractical. Hence, reliance on sump cleaning agencies or individuals is necessary.

Generate solutions for cleaning the sump using the analogies provided. Residents should not depend on external agencies to remove the dirty water in the sump.

We present a detailed description of the problem to benefit the international readership. A sump, commonly found in India, refers to an underground or partly submerged tank. Traditionally constructed with bricks or reinforced concrete, sumps are integral components of rainwater harvesting systems or used for storing groundwater, which is pumped into overhead tanks [35]. Cleaning sumps is challenging as the water cannot be completely drained out, and their ground-level placement exposes them to dirt and insect infiltration.

We chose this problem for several reasons: (a) ensuring clean water and good health are two sustainable development goals [36]; (b) the participants, who were Indians, had a detailed understanding of the problem due to its prevalence in India; and (c) the problem falls within the domain of product design—the area of specialization of the participants.

Analogies: Participants were given analogies from the three domains to stimulate idea generation. See Figure 1 for the biological-domain analogy drawn from the Pitcher Plant; Figure 2 for the cross-domain analogy, inspired by the Backhoe Loader; and Figure 3 for the within-domain analogy, drawn from the Vacuum Cleaner. The criteria for selecting these analogies were based on the following author’s judgment: (a) whether the function of the analogy aligned with the desired action of the selected problem, and (b) whether the selected analogy could be transferred in the stipulated time.

As depicted in Figure 1, Figure 2 and Figure 3, each analogy was represented using text accompanied by two images. A similar approach to the representation of analogies was used by [37].

The textual description of the biological domain analogy was obtained from Asknature.org [38] and the descriptions of cross-domain and within-domain analogies were taken from HowStuffWorks.com? [39]. We chose the descriptions of analogies from these sources because AskNature is a scientifically validated tool for ideation using biological inspiration [25], and HowStuffWorks.com [39] is a well-known encyclopedia providing descriptions of technical systems, which was previously used to create a database of tools supporting the DbA method [40]. We ensured that the descriptions of selected analogies were sufficiently comprehensive to allow designers to understand the workings of the analogous system. This mitigated bias in our results arising from variations in the level of detail at which an analogy was described. The number of words for each of the biological-, cross-, and within-domain analogies was roughly similar—162, 202, and 212 words, respectively. While it was impossible to maintain an identical word count due to differences in the complexity of each analogous system, we tried to achieve a comparable level of detail in the descriptions.

Two images accompanied each textual description of the analogies. However, for the biological analogy, the images represented the form of the system. In contrast, one image showed the system’s form for the other analogies, and the other depicted a block diagram. We opted not to include block diagrams for the biological-domain analogies because, based on our judgment, the block diagrams of pitcher plants might be challenging to interpret by the engineering-oriented designers. The participants were also given blank A4 sheets, pens, and pencils to sketch their design ideas.

**Figure 1 biomimetics-09-00344-f001:**
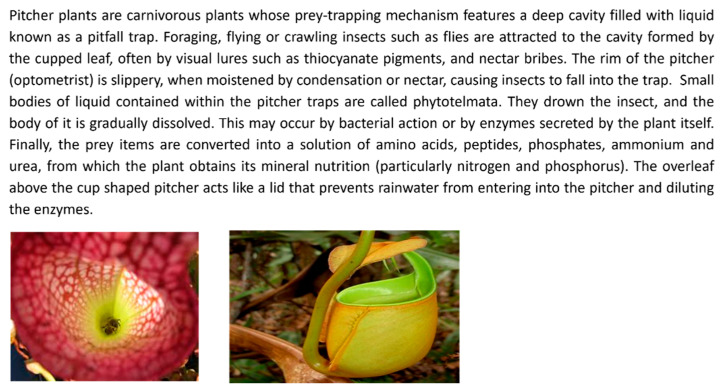
**Biological-domain analogy**—How does a pitcher plant catch insects and decompose them? [38].

**Figure 2 biomimetics-09-00344-f002:**
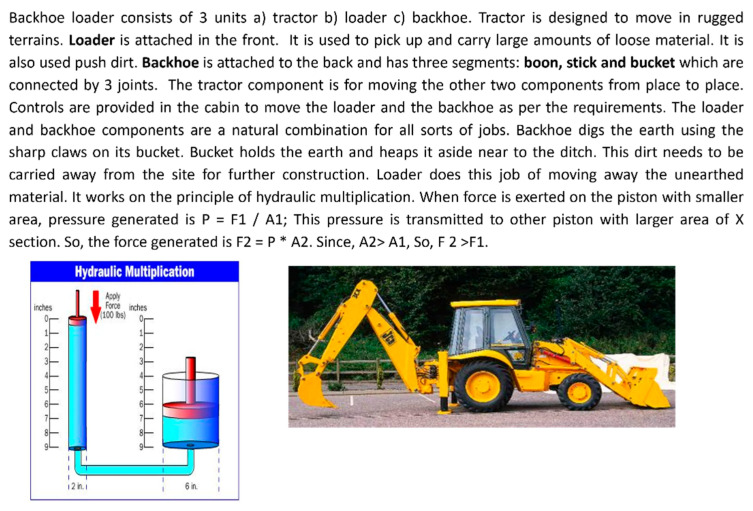
Cross-domain analogy—How does a backhoe loader dig the earth? [39].

**Figure 3 biomimetics-09-00344-f003:**
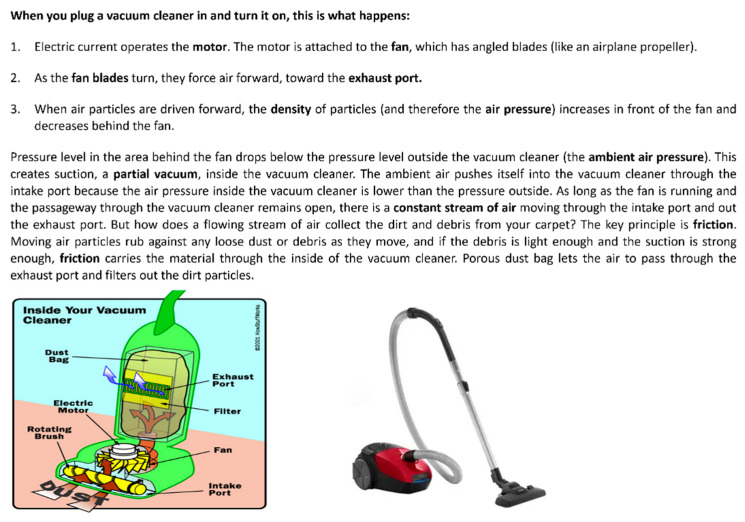
Within-domain analogy—How does a vacuum cleaner clean things? [39].

## 3. Analysis of Design Ideas

This section presents the units of analysis of design ideas and examples of ideas produced in the experiment.

Overall, the teams generated a total of 148 concepts in our experiment. Of these, 13 concepts were generated using more than one analogy. This contradicted the instructions given to the subjects. Consequently, we excluded these concepts from our analysis. The remaining 136 concepts were analyzed based on the following units of analysis:**Frequency of Selection** of analogy domain: This unit allowed us to test Hypothesis H1. It represented idea fluency from a particular domain. Subjects were instructed to mark the concepts chronologically and provide the name of the analogy that inspired that idea (see instruction 3 in Appendix A). We used these markings to calculate the frequency of analogy domain selection.**Novelty of Ideas Produced**: This unit addressed Hypothesis H2 and employs the method proposed by Sarkar and Chakrabarti [4] to evaluate the novelty of concepts. The novelty evaluation method uses the SAPPhIRE model of causality [41,42], an acronym encompassing the constructs of **S**tate–**A**ction–**P**art–**Ph**enomena–**I**nput–**O**rgan–**E**ffect. This model is described as follows [43]:

*Parts* consist of *entity and surroundings*. An *entity* is a subset of the universe under consideration and is characterized by its boundary; *surroundings* include all the subsets of the universe except for the entity. Components and interfaces that comprise an entity and its surroundings (i.e., parts) have various properties and conditions (*organs*). When the entity and the surroundings are not in equilibrium, physical quantities are transferred in the form of material, energy, or signal (*input*) across the entity boundary. These inputs, in combination with organs, activate principles (*physical effects*). Physical effects are responsible for interactions (physical phenomena) between the entity and its surroundings. The interaction changes various properties of the entity and the surroundings (*state-change*). The changes in properties can be interpreted at a higher level of abstraction (*action*). Figure 4 illustrates the logical dependency among the constructs in the SAPPhIRE model, and the digits 1–7 in parenthesis denote the hierarchy in the SAPPhIRE abstraction levels.

Figure 5 shows the steps for evaluating the novelty of ideas [4]. Moss-Metric [44] and Sarkar and Chakrabarti’s method [45] were recommended for assessing the novelty of design ideas due to their strong agreement with expert evaluations [46]. The novelty assessment method [4] used in this study builds upon the method proposed by Sarkar and Chakrabarti [45]. Further, our work builds upon the work of Keshwani et al. [16], who used Sarkar and Chakrabarti’s method [4] for novelty evaluation. Consequently, we used the method of Sarkar and Chakrabarti [4] for our novelty evaluation.

To conduct this assessment, the authors created a comprehensive database of existing ideas through Internet research (see Appendix B). After that, SAPPhIRE models were constructed and developed for both—the ideas produced in the design sessions and pre-existing ones. These models were then compared using the method described in Figure 5.

Each idea was then categorized into No Novelty, Low Novelty, Medium Novelty, High Novelty, and Very High Novelty (See Figure 5). The novelty categories were further converted into novelty scores using Table 4, adopted from the novelty evaluation metric proposed by Srinivasan and Chakrabarti [42].

The first author of this paper carried out the novelty assessment. She assessed a similar novelty for nine other empirical studies [16,46]. In four of these empirical studies, an inter-encoder reliability test for novelty assessment was carried out using encoders with about five years of experience in the SAPPhIRE model and in evaluating novelty [47]. This resulted in a 90% rate of agreement (see Appendix B of [47]). This exceeds the generally accepted threshold of 70% for inter-encoder reliability [48]. Since the author previously conducted an inter-encoder reliability study using the same method as this study, no separate test was performed for this research.

Figure 6a–c illustrate the ideas generated using biological-, cross-, and within-domain analogies. In Figure 6a, the designers propose keeping the inlet of the sump slightly above the ground to avoid the entry of unclean water from the sides. The sump inlet is covered with an airtight lid with a cap. The lid has a mesh attached to trap the insects or dirt that can enter through the inlet. If the trap fails to catch the insects or dirt, the users can induce chemicals in the water through the cap. These chemicals can convert the insects into harmless proteins. In Figure 6b, the designers propose having a portable piston-like attachment connected to the bottom of the sump through a one-way valve. When the piston is moved upward, dirty water and settled insects at the bottom of the sump will be sucked through the one-way valve and removed from the other end. The designers rely on gravity to settle the dirty water at the bottom of the sump. In Figure 6c, the designers propose adding an extra attachment to the existing vacuum cleaner. This attachment can be lowered into the sump and has a wide opening with holes that can suck the dirty water collected at the bottom of the sump.

## 4. Results

This section presents the experiment results for testing Hypothesis H1 (see Section 4.1) and Hypothesis H2 (see Section 4.2). As the samples were paired and the observations were not found to be normally distributed, we used non-parametric statistical tests—Wilcoxon Signed-Rank test [49] for testing H1, and Spearman’s Rank Correlation test [49] and Fisher Z-Transformation [49] for testing H2.

### 4.1. Testing Hypothesis H1: f_cross_ > f_within_ > f_bio_

Overall, out of 136 ideas, the proportion of selection of biological-, cross-, and within-domain analogies were 45.5%, 29.4%, and 25%, respectively. Figure 7 shows the distribution of the Number of Ideas produced by each team (A–Z) using biological-, cross-, and within-domain analogies; 13 out of 26 teams (50%) produced at least 50% of total ideas using biological-domain analogies. Figure 8 shows the scatter plot for the total number of teams against the number of ideas they produced using each analogy domain. The maximum number of ideas generated using cross- and within-domain analogies was 3, while the maximum number of ideas produced using the analogies from the biological domain was 7.

Overall, these results indicate that the designers select biological analogies the most over cross-domain and within-domain analogies.

To test Hypothesis H1, we performed a pairwise comparison across the domains using the Wilcoxon Signed-Rank Test [49] calculated at the α = 0.05 level of significance for the two-tailed test. See Table 5 for results. The frequency of selection of biological-domain analogies was significantly higher than that of cross-domain analogies (*p* < 0.05) and within-domain analogies (*p* < 0.05). On the contrary, no significant difference was found between the frequency of selection of cross- and within-domain analogies. Thus, our results do not support Hypothesis H1.

### 4.2. Testing Hypothesis H2: r_bio_ > r_cross_ > r_within_

Figure 9 shows the novelty variation of ideas produced using each analogy domain across teams A–Z. Overall, 10 (38%) teams produced more than 50% of novelty using a biological-domain analogy, four teams (15.3%) produced more than 50% of novelty using a cross-domain analogy, and one team (3.8%) produced more than 50% of novelty using a within-domain analogy. The overall percentage distribution of novelty, across teams A–Z, for all the ideas created by biological-, cross-, and within-domain analogies was found to be 47%, 32.4%, and 20.6%, respectively.

To test Hypothesis H2, for each analogy domain, the Spearman Rank Correlation was calculated between the two parameters—Frequency of Selection of Analogy Domain (f_d_) by a team and the Overall Novelty of Ideas (n) produced by the team. The results indicated that, except for the within-domain analogies, for all other cases, there is a significant and strong positive correlation [50] (Cohen [50] proposed the following limits for Strength of Correlation: Weak correlation: |r| < 0.30, Moderate correlation: 0.30 ≤ |r| < 0.50, and Strong correlation: |r| ≥ 0.50) between the two parameters f_d_ and n (see Table 6).

Although r_bio_ > r_cross_, upon applying the Fischer Z-Transformation [49], no significant difference was found between these correlation coefficients. Thus, we did not find strong evidence in favor of Hypothesis H2.

Table 7 presents the Wilcoxon Signed-Rank Test results for pairwise comparisons between the three analogy domains. The analyses revealed weak evidence indicating a significant difference in the novelty of ideas between the two pairs: biological- vs. cross-domain and cross- vs. within-domain analogy. However, a significant difference was observed between the novelty of ideas generated using biological- and within-domain analogies (*p* < 0.05).

## 5. Discussion

The following are the main results of this work:The Frequency of Selection of biological-domain analogies was significantly higher than that of cross-domain and within-domain analogies.The Frequency of Selection for cross- and within-domain analogies was not found to be significantly different.For biological- and cross-domain analogies, the correlation coefficient between Frequency of Selection and Novelty of Ideas was strongly positive and statistically significant.For within-domain analogies, a non-significant weak positive correlation was observed between the Frequency of Selection and Novelty of Design Ideas.While numerically, we observed the trend r_bio_ > r_cross_ > r_in_. However, no significant differences were found among the three correlation coefficients.The Novelty of Ideas produced using biological analogies was found to be significantly higher than that of within-domain analogies.The Novelty of Ideas produced using cross- and within-domain analogies, as well as between biological- and cross-domain analogies, were not found to be significantly different.

We present the interpretation of these results in the context of research questions RQ1 and RQ2. Regarding the first research question, about the relationship between analogy domains and their Frequency of Selection (f_d_) during the design process, we found the following:

Firstly, our findings (a and b above) indicated that, when dealing with the design problem, participants generally selected biological analogies over cross- and within-domain analogies. These results do not support Hypothesis H1—that cross-domain analogies would produce the most novel designs. Furthermore, our results contradict Ward’s proposition of the path of least resistance [20] and differ from previous studies [26,27]. While these two studies [26,27] suggest that designers do not prefer far-domain analogies, in our research, designers—who had 2–3 three years of experience—somewhat similar to that of [27], chose biological domain analogies—the farthest-domain analogy—over cross- and within-domain analogies. A possible reason for this difference may be that, in our work, the farthest-domain analogies belonged to the Biology discipline, whereas, in these studies, the farthest-domain analogies were in the Engineering discipline. The participants might have been attracted to use analogies from the Biology discipline (biological-domain analogies) rather than the analogies from the Engineering discipline (cross- and within-domain analogies). We propose the following reason for this result: According to Berlyne’s [51] classic study on curiosity, individuals exhibit more curiosity when they encounter stimuli that are unfamiliar or complex, as these characteristics generate a state of uncertainty that motivates exploration and information-seeking. Similarly, a study on the neural mechanisms underlying curiosity suggested that complex or uncertain stimuli activate the brain’s reward mechanism, increasing motivation and engagement in learning and problem-solving [52]. In our study, the participants were engineers. To them, the biological analogy might have been the most different and complex analogy from the other two. This might have produced greater curiosity, motivation, and engagement towards using biological-domain analogies.

Secondly, as the frequency of selection of the analogy domain also represents idea fluency, our results show that biological-domain analogies positively influence the fluency of ideas. This is a novel contribution since, to our knowledge, no study has compared the influence of biological- and engineering-domain analogies in terms of the fluency of ideas.

Regarding RQ2, dealing with the relationship between the Frequency of selecting analogies from specific domains (f_d_) and the Novelty of the Designs (n), we found the following:

Firstly, findings (c–e) showed a robust correlation between the frequency of selection and the novelty of designs for biological- and cross-domain analogies, indicating a strong link between idea fluency and the generated novelty. This suggests that biological- and cross-domain analogies are more promising in inspiring innovative designs than within-domain ones. Although our analysis showed a numerical hierarchy (r_bio_ > r_cross_ > r_within_), we did not observe a statistically significant difference among the three correlation coefficients. Thus, Hypothesis H2 lacks strong support. This might be partly due to our study’s limited number of observations about the availability of participants with the desired educational background.

Secondly, findings (f–g) support the results (c–e), indicating that the greater the conceptual distance between source and target domains, as observed in biological-domain analogies, the higher the novelty of the design outcomes, compared to the situation where these two domains are conceptually closer, as in the case of within-domain analogies. However, our study did not show significant differences between the cross- and biological-domain analogies regarding the novelty of ideas generated. These findings are consistent with those of Keshwani et al. [16], who also classified analogies as belonging to biological, cross, and in (or within) domains despite using different problems and types of analogies. Additionally, our results suggest that the farthest-domain analogies (biological-domain) may not necessarily yield the most novel designs, implying that middle-domain analogies could be as effective as far-domain analogies in supporting novel idea generation. These align with prior studies [13,14,17]. Ruiz-Pastor et al. [29] reported no significant difference in the novelty of ideas produced using biomimetic and random stimuli. However, in our view, our results cannot be compared to their results because, while there is a clear distinction between cross-and within-domain analogies in our work, on the contrary, the random images in their work [29] seem to be a combination of cross- and within-domain analogies. For example, for the problem ‘design of innovative outdoor element for refuge,’ the image of a building (Image 3) could belong to a within-domain analogy and the image of an electronic device (Image 9) could belong to a cross-domain analogy. Future work could classify the random images into two domains—within- and cross-domain, and study the individual influence of each domain on the novelty of concepts of the produced design outcomes.

While biological-domain analogies did not contribute significantly to generating the most novel designs, designers with an engineering background showed a more significant ability to establish analogical correspondences with cross-domain sources than biological-domain ones. Therefore, the results suggest that, although designers preferred biological analogies—as evidenced by their increased selection frequency—their lack of knowledge in the biology discipline may have affected their ability to effectively transfer analogical principles from this source to the design problem.

It can be argued that the design problem influenced our results. However, Keshwani et al. [16], who proposed the classification scheme used in our study, conducted experiments with three design problems and obtained results consistent with our work. Hence, our results do not seem to be influenced by the design problem.

This work contributes significantly to the design field regarding the use of analogies; however, a potential limitation is the short time allocated for carrying out the design task. Further studies could extend this work to investigate the impact of time on the selection of analogy domains and the novelty of the designs. Current results might vary if the designers who participated in our study had more experience using biological analogies. It is speculated that, with a higher level of expertise, the frequency of selection of biological domain analogies may decrease because it will no longer represent a challenge to them. On the other hand, the novelty of the produced biological-domain-based concepts may probably increase due to their developed cognitive capability to transfer and apply biological principles to engineering problems.

Moreover, future research could test the findings of this study using alternative novelty assessment methods, such as [5,44]. Exploring the relationship between the analogy domain and the time designers spend using each analogy domain could offer important insights. Likewise, investigating the role and attributes of analogy representation modalities, such as images and videos, in selecting analogy domains could provide additional opportunities for exploration in future studies. Similarly, investigating the attributes of designers, such as their experience and level of understanding in selecting and using analogies, is an interesting future work.

This work has applications for tools for the retrieval of analogies (for example, see [25,41]). For the development of these tools, recent research efforts are directed towards increasing the number of analogies retrieved by the designers (for example, see [40,53,54]). As the number of retrieved analogies will increase in these tools, they must present them in decreasing order of their potential to inspire novel ideas. Results c, d, and e at the beginning of Section 5 suggest that the biological- and cross-domain analogies have more potential for producing novel ideas than the within-domain ones. Therefore, these tools must present the retrieved analogies to the designers in the following order: (1) biological analogies; (2) cross-domain analogies, and (3) within-domain analogies at the end.

## 6. Conclusions

In this study, we used an empirical approach to explore the selection criterion of analogy domains and the relationship between the frequency of selection of analogy domains and the novelty of ideas. Our findings indicated a strong preference for designers to select biological analogies over those from far- and near-engineering domains. Additionally, we observed a significant positive influence of both biological- and cross-domain (far-engineering domain) analogies on the novelty of designs. However, we were unable to find which domain produces more novel designs. Despite designers’ preference for biological analogies over analogies from the other domains, we speculate that they found applying them to generate novel designs challenging. The main contributions of this study are as follows:Previous studies have compared the influence of individual analogy domain(s) on novelty designs [8,15,18,19,20]. In contrast, our study compared this influence using analogies from various domains—a scenario closer to real-world analogy use.The selection of biological analogies indicates designers’ preference for bio-inspired design over the Design-by-Analogy (DbA) method. This suggests that designers perceive a more significant potential for biological analogies than analogies from other domains.To our knowledge, previous studies have not reported the influence of analogy domains on the fluency of design ideas. This study addressed this gap and found that idea fluency is highest for biological-domain analogies.Previous studies have classified analogy domains into a tripartite scheme—far-, middle-, and near-domains [13,14,17]—all within the Engineering discipline. However, we included analogies from biological and engineering disciplines in our classification of analogy domains. This broadened the scope and compared Biologically Inspired Design and Design-by-Analogy methods regarding their influence on analogy domain selection and novelty.While there have been studies on the influence of the DbA method on novelty, to our knowledge, none of the previous studies have compared correlation coefficients between fluency and novelty of ideas for biological-, cross-, and within-domain analogies—a comparison we conducted in this work.Our work considers analogies from both the biological and engineering domains. Therefore, our findings can be relevant for developing tools such as Idea Inspire [55] that have databases of analogies from the biological and engineering domains—the two domains considered in this work. Such tools can present the retrieved analogies to the designers in the following order: (1) biological analogies; (2) cross-domain analogies, and (3) within-domain analogies at the end.

## Figures and Tables

**Figure 4 biomimetics-09-00344-f004:**
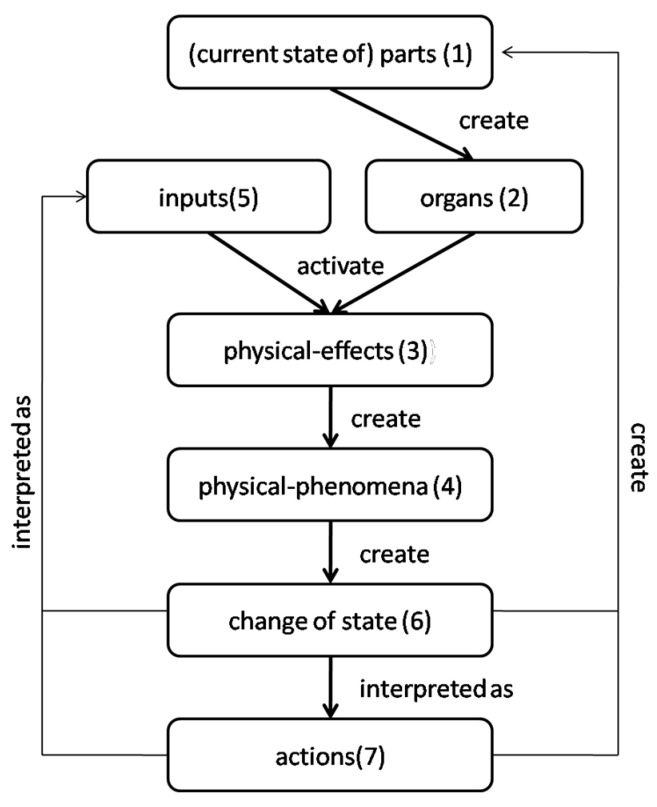
The SAPPhIRE model of causality [40,41].

**Figure 5 biomimetics-09-00344-f005:**
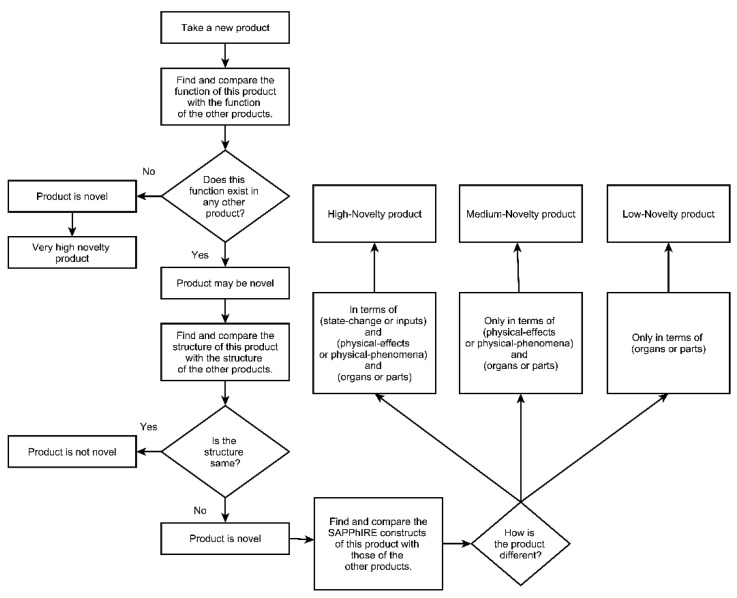
Steps to identify the novelty of products [4].

**Figure 6 biomimetics-09-00344-f006:**
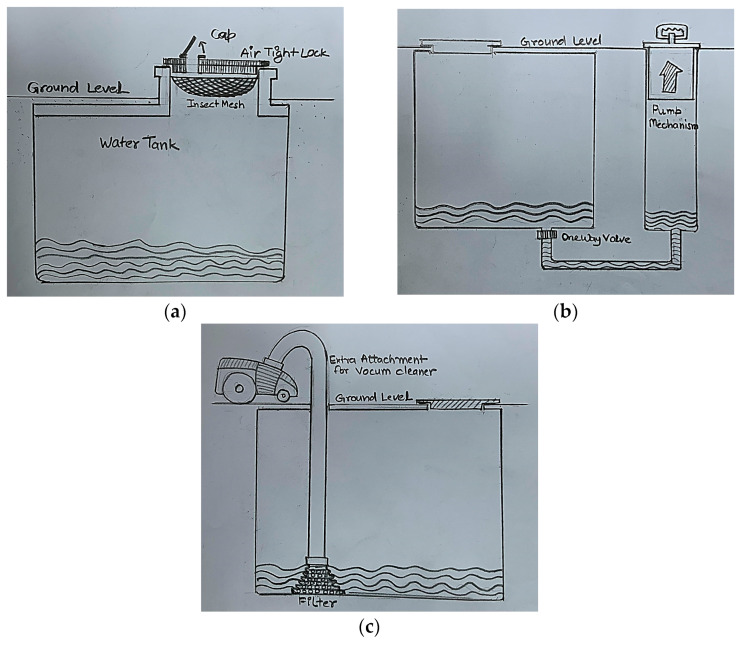
(**a**) An example idea generated using biological-domain analogy; (**b**) an example idea generated using cross-domain analogy; and (**c**) an example idea generated using within-domain analogy.

**Figure 7 biomimetics-09-00344-f007:**
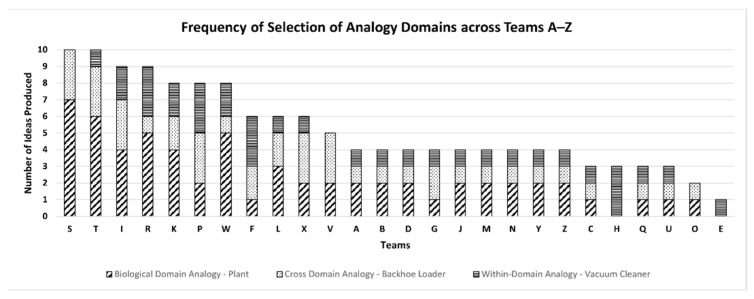
Distribution of the Number of Ideas produced by each team using biological-, cross-, and within-domain analogy.

**Figure 8 biomimetics-09-00344-f008:**
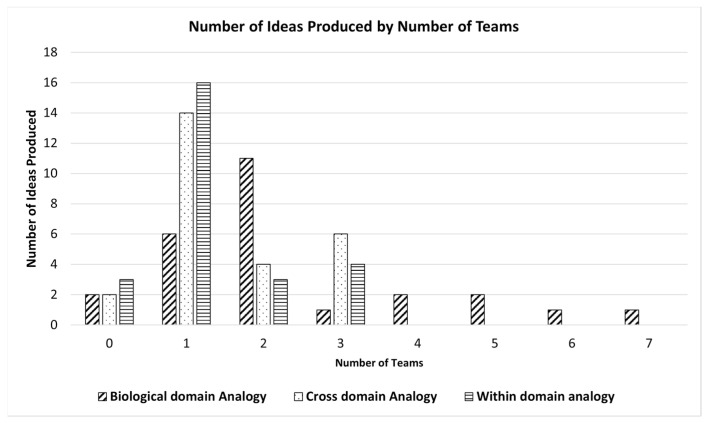
Number of Ideas produced by the teams according to analogy domain.

**Figure 9 biomimetics-09-00344-f009:**
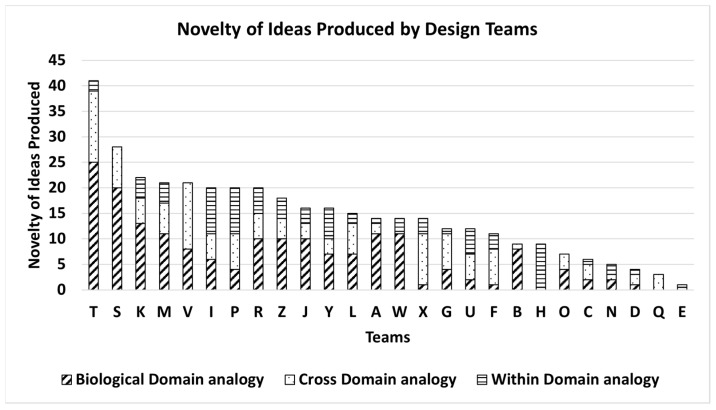
Distribution of the Novelty of Ideas produced by teams A–Z according to analogy domain.

**Table 1 biomimetics-09-00344-t001:** Classification schemes of analogy domains proposed in literature.

Domain Classification Schemes	References	Description	Influence on Novelty
Near- or Within-domain and Far- or Between-domain	[8,9,15,18,19]	**Near-domain analogies:** The source and target domains are conceptually very close. **Middle-domain analogies:** When the source and the target domains are neither too far nor too close but somewhere in the middle. **Far-domain analogies:** When the source and the target domains are conceptually different.	Far-domain analogies produced more novel designs than near-domain analogies.
Near-, Middle-, and Distant-domain	[13,14,17]		The largest positive influence on the novelty of design outcomes was when the conceptual distance was neither too high nor too low, i.e., somewhere in the middle.
Engineering-domain and Biological-domain	[21,23]	**Engineering-domain analogies** represent man-made technical systems. **Biological-domain analogies** represent natural systems	Biological-domain analogies produced more novel designs than engineering-domain analogies.
In-domain, Cross-domain, and Biological-domain	[16]	**In-domain (near-engineering domain) analogies:** When the source domain of the analogy is similar to that in which the problem (target) is to be solved. **Cross-domain (distant-engineering domain) analogies:** When the source domain of the analogy is different from that in which the problem (target) is to be solved. **Biological-domain analogies:** When the source domain of the analogy belongs to an organism or its interaction with the environment.	Cross-domain and biological-domain analogies produced significantly more novel designs than in-domain analogies. No significant difference was reported between biological- and cross-domain analogies.

**Table 2 biomimetics-09-00344-t002:** Examples of near-, middle-, and far-domain analogies for the design task—‘Proposed Ideas for Next-Generation Cleaning Robots’ [13].

Analogy	Example
**Near-domain analogy**	**Text:** Dust particles are adsorbed through electrostatic force and fall into the collection plates after losing electricity. **Function:** Clean dust. **Context:** Domestic use (same context as the vacuum cleaner).
**Middle-domain analogy**	**Text:** The stress that bondless windscreen wipers support is distributed. They fit with glass perfectly, reducing the damage and eliminating the vibration caused by wind. **Function:** Clean dust. **Context:** Outdoor use/road infrastructure.
**Far-domain analogy**	**Text:** Magnetic separation can extract magnetic material from a mixture through a magnetic field applied in mining iron and garbage classification contexts. **Function:** Separate solids (different but relevant function). **Context:** Industrial use.

**Table 4 biomimetics-09-00344-t004:** Novelty Category of an idea is converted into its Novelty Score using the metric proposed by [42].

Novelty Category	Construct at Which the Two SAPPhIRE Models Differ	Novelty Score
No Novelty	No difference between the two SAPPhIRE models	0
Low Novelty	Part	1
	Organ	2
Medium Novelty	Physical-Effect	3
	Phenomena	4
High Novelty	Input	5
	State-Change	6
Very High Novelty	Action	7

**Table 5 biomimetics-09-00344-t005:** Results of the pairwise comparison for frequency of selection across biological-, cross-, and within-domain analogies (f_bio_ = 62, f_cross_ = 40, f_within_ = 34).

Pairwise Comparison between Domains	*p*	Remarks
Biological-Domain vs. Cross-Domain	0.011	*p* < 0.05, f_bio_ > f_cross_
Cross-Domain vs. Within-Domain	0.392	*p* > 0.05, f_cross_ ~ f_within_
Within-Domain Vs. Biological-Domain	0.009	*p* < 0.05, f_within_ < f_bio_

**Table 6 biomimetics-09-00344-t006:** Correlation between Frequency of Analogy Domain Selection and Novelty of Design Ideas.

Analogy Domain d	Spearman’s Rank Correlation Coefficient between f_d_ and n	Remarks
Biological-Domain	r_bio_ = 0.73, *p* < 0.001	Significant and strong positive relationship between f_bio_ and n.
Cross-Domain	r_cross_ = 0.62, *p* < 0.001	Significant and strong positive relationship between f_cross_ and n.
Within-Domain	r_within_ = 0.0516, *p* = 0.802	Non-significant and weak positive relationship between f_within_ and n.

**Table 7 biomimetics-09-00344-t007:** Results of Pairwise Comparison of Novelty of Ideas produced using biological-, cross-, and within-domain analogies (n_bio_ = 178, n_cross_ = 123_,_ n_within_ = 78).

Pairwise Comparison between Domains in Terms of Novelty of Designs	Significance Level *p*	Remarks
Biological-Domain vs. Cross-Domain	0.088	n_bio_ ~ n_cross_
Cross-Domain vs. Within-Domain	0.084	n_cross_ ~ n_within_
Within-Domain Vs. Biological-Domain	0.012	n_within_ < n_bio_

## Data Availability

Data can be made available upon request by contacting to sonalkeshwani@gmail.com.

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
