# Peer review of "Comparing Analogy-Based Methods—Bio-Inspiration and Engineering-Domain Inspiration for Domain Selection and Novelty"

_biomimetics, 2024, doi:10.3390/biomimetics9060344_

Round 1

Reviewer 1 Report

Comments and Suggestions for Authors

The study investigates the impact of selecting analogies from far, near, and middle domains on problem-solving, and explores the relationship between the frequency of selecting analogies from these domains and the novelty of the resulting design solutions. Far-domain analogies are those conceptually distinct from the target domain, while near-domain analogies share similarities with the target domain. Middle-domain analogies fall between these two extremes. Through a study involving twenty-six teams of novice product designers, the research investigates how biological, cross-domain, and near-domain analogies influence selection frequency and the novelty of designs generated. Results revealed a high consideration for biological analogies, with no significant difference in selection frequency between cross-domain and near-domain analogies. Notably, biological analogies enhanced the novelty of designs compared to near-domain analogies.

This study offers valuable insights into the potential of biological and cross-domain analogies in generating innovative solutions to industrial challenges, although certain aspects of the research need further clarification.

Section 1: Introduction

a)      Lines 49-52: “Based on the findings, these tools can be modified to prioritize the search results from those analogy domains which are more likely to be selected and are likely to produce more novel designs. This in turn, will support technological innovation.”

Questions: 1)      Does this mean that when a designer searches for a solution to a problem, the tool will first retrieve biological solutions and then be followed by middle domain and within-domain solutions?

2)      Tools such as AskNature already have a biological database where a designer can search for an already existing biological solution to a problem. How do these findings contribute to developing tools?

b)      Lines 97-99: “No work has investigated analogy selection from biological domain.”

Question: 1)      Although the author points out a significant growth in the biological domain by citing some recent works, findings from the following article reveal that biomimicry aids in obtaining creative, novel, and circular products. 

“Ruiz-Pastor, L. et al. (2023) ‘Bio-inspired design as a solution to generate creative and circular product concepts’, International Journal of Design Creativity and Innovation, 11(1), pp. 42–61. doi: 10.1080/21650349.2022.2128886.”

The authors are requested to elaborate on the difference between such findings and this research.

Section 2: Materials and Methods

a)      Figure 5: Evaluating the novelty of the generated products.

Question:

1)      Was a literature search performed to find existing products that perform the same function as the ones that are generated?

Section 4: Discussion

a)      Lines: 444-449: “greater the conceptual distance between the source- and target-domains, as observed in biological domain analogies, the higher the novelty of the design outcomes, compared to the situation where these two domains are conceptually closer, as in the case of within-domain analogies. However, our study did not show significant differences between the cross- and biological -domain analogies regarding the novelty of the ideas generated.”

And Lines: 456-457: “While the use of biological-domain analogies did not contribute significantly to generating the most novel designs,”

Question:

1)      Although the author elaborates on the reason for novice designers choosing biological analogies over cross and within-domain analogies in the following section, the question here is: what if this empirical study is performed on designers who have experience in bio-inspired design?

2)      Will the results show a significant increase in the novelty score on biological-domain analogies?

3)       Can we say that the results of such empirical studies vary with the experience and level of understanding of the designers?

Section 5: Conclusions

a)      Lines 513-515: “The criteria for selecting analogies can be employed to populate the database according to the preferred analogy domains and rank the search results accordingly.”

Question:

1)      Do the criteria here imply that the frequency of selection and novelty for the biological domain is always higher? Please define the criteria.

Comments on the Quality of English Language

Please make sure to double check the English of the manuscript. 

Reviewer 2 Report

Comments and Suggestions for Authors

This interesting paper is worth of publication. I suggest a few improvements to be made.

1 - In section 2.2 Subjects:

Could you better characterize the subjects, not only in terms of age, gender and training background but also from cultural and geographical origins (country and places of origin, as well as urban as or countryside background)?

Where are the subjects from? The same country and culture? From where? Was the experiment conducted in only one city? What was the origin (city) of each individual subject? In the materials sections it is said that the participants are Indians.

2 - In section 2.4 Materials

Asknature.com is Asknature.org. It is in the in line 206 at page 6 and in the description of figure 1.

HowStuffWorks can be addressed as HowStuffWorks.com.

3 - Figure 3:

the scheme (taken from https://home.howstuffworks.com/vacuum-cleaner.htm) and the photograph have a different sequence of components: in the scheme the motor and the fan blades are placed before the collecting bag, unlike the house-used vacuum cleaner of the photograph, although that might no be relevant to the purpose it was intended to.

4 - Sub section 2.5 Analysis of Design Ideas

Does this subsection belong here or should be moved to the results and analysis section?

5 - The graphic on figure 7: the results are presented according to the sorting of the teams (A to Z), which conveys no information at all. If, however, the data were sorted from the higher number of ideas to the lowest, that would me meaningful. It is also suggested that the different filling shades of gray in the bars should be replaced or complemented with different filling patterns to better distinguish the results, or else, using dark, gray and white. The same remarks apply to figure 9.

6 - The readability of figure 8 is also affected by the color and the size of the symbols chosen. 

7 - Fill and replace the instructions provided in the template for:

Author Contributions, Funding, Institutional Review Board Statement, and Data Availability Statement.
